# Telemedicine adoption in cardiology: Determinants and predictors identified using Bayesian Model Averaging and Machine Learning

Pascal Petit[1☯], Jonathan Nübel[2,3☯], Marie Josephine Walter[2,3], Christian Butter[2,3], Martin Heinze[4], Yuriy Ignatyev[4], Anja Haase-Fielitz[2,3,5], Nicolas Vuillerme[1,6‡], Felix Muehlensiepen[1,3,7*‡]

1 Institute of Engineering, Univ. Grenoble Alpes, CNRS, Grenoble INP, LIG, SANGRIA, Grenoble, France, 2 Department of Cardiology, University Hospital Heart Centre Brandenburg, Brandenburg Medical School Theodor Fontane, Bernau/Neuruppin, Germany, 3 Faculty of Health Sciences, Joint Faculty of the University of Potsdam, the Brandenburg Medical School Theodor Fontane, and the Brandenburg Technical University Cottbus-Senftenberg, Cottbus, Germany, 4 Department for Psychiatry and Psychotherapy, Center for Mental Health, Immanuel Hospital Rüdersdorf, Brandenburg Medical School Theodor Fontane, Rüdersdorf, Germany, 5 Institute of Social Medicine and Health System Research, Otto von Guericke University Magdeburg, Magdeburg, Germany, 6 Institut Universitaire de France, Paris, France, 7 Center for Health Services Research, Faculty for Health Sciences, Brandenburg Medical School Theodor Fontane, Rüdersdorf, Germany

☯ These authors contributed equally and share the first authorship on this work.
‡ NV and FM also contributed equally and share the last authorship on this work.
* Felix.Muehlensiepen@mhb-fontane.de

## Abstract

In this secondary analysis of a German cross-sectional survey data, we investigated key determinants and predictors of telemedicine (TM) use among healthcare professionals (HCPs) treating cardiology patients. We applied Bayesian Model Averaging (BMA) for explanatory analysis and Machine Learning (ML) for predictive modeling. BMA identified TM determinants after excluding collinear variables and selecting variables based on LASSO regression. The extreme gradient boosting (XGBoost) ML algorithm predicted TM use and identified key predictors, using nested cross-validation to prevent overfitting. ML model performance was assessed via area under the receiver operating characteristic curve (AUROC), while predictor importance was evaluated using Shapley additive explanations. Among 112 HCPs, 64 (57%) used TM. BMA identified 12 determinants, including positive associations with TM knowledge, being a cardiologist, female gender, and perceiving TM as suitable for heart failure and for monitoring events. Negative associations included concerns about insufficient patient benefits, perceptions that TM is less suitable for acute events, and skepticism regarding its relevance for extending aftercare intervals. The XGBoost model showed strong predictive performance (AUROC: 0.88 [95% CI: 0.75; 1.00], accuracy: 0.79) for TM use. Key promoting factors included TM knowledge, being a cardiologist, female gender, number of average patients per quarter, and perceiving TM as suitable for arrhythmias, device follow-up, and heart failure. Limiting factors

**Data availability statement:** The anonymized survey dataset underlying the findings of this study, including the participant data and data dictionary, is provided in the Supporting information files. The codes used in the analysis of this study is publicly available at: https://doi.org/10.5281/zenodo.17047856.

**Funding:** The publication fee was covered by the Brandenburg Medical School publication fund supported by the Ministry of Science, Research and Cultural Affairs of the State of Brandenburg. The work of PP and NV was has been supported by funding from the French government, managed by the French National Research Agency (ANR) under the France 2030 program (MIAI Cluster) [ANR-23-IACL-0006] and under the "Investissements d'avenir" program [ANR-10-AIRT-0005; ANR-15-IDEX-0002].

**Competing interests:** The authors have declared that no competing interests exist.

included older age, personal use of TM for one's own health, and skepticism about TM's relevance in acute situations. These findings emphasize the importance of knowledge and attitudes in shaping TM adoption and show that ML can accurately identify healthcare professionals most likely to use TM, supporting targeted interventions and safer implementation in cardiology.

## Author summary

In our study, we wanted to better understand why some healthcare professionals in Germany use telemedicine for cardiology patients, while others do not. We surveyed doctors about their experiences and opinions on using digital tools in heart care. To understand better what drives or hindered participants we used two types of analysis: one that finds important influencing factors, and one that predicts which professionals are most likely to use telemedicine. We found that healthcare professionals who had strong knowledge of telemedicine and who perceived it as relevant for patient care were much more likely to adopt it. Being a cardiologist and self-identifying as female were also consistently associated with higher use. In contrast, those who expressed doubts about its patient benefits, considered TM unsuitable for acute events, or reported insufficient knowledge were less likely to engage with it. Our predictive model achieved high accuracy in identifying those most likely to adopt telemedicine. While these results may appear intuitive, our use of advanced statistical and machine learning methods provides robust, data-driven evidence for patterns that often seem self-evident in practice. By quantifying these associations, our work offers a stronger foundation for healthcare policy, such as promoting education on telemedicine and addressing barriers for those who feel less confident using digital tools.

## Introduction

The digital transformation of healthcare is advancing rapidly, with cardiology historically serving as a key driver of medical innovation [1]. From the early adoption of the electrocardiogram (ECG) as one of the first digital diagnostic tools [2] to the widespread implementation of remote monitoring [3] and AI-assisted diagnostics [4], cardiology has consistently embraced and shaped technological advancements to improve patient care [5].

With an aging population and a rising prevalence of cardiovascular disease [6], the demand for cardiology services is increasing even as the shortage of healthcare professionals intensifies [7]. Digital health technologies, particularly telemedicine (TM), offer a promising solution to address this growing healthcare challenge [8–13]. TM encompasses a wide range of applications, from remote consultations to continuous telemonitoring, offering diverse solutions for patient care and clinical decision-making. The American Heart Association has underscored these advantages, yet significant

challenges, such as insufficient technological infrastructure, reimbursement issues, and limited digital literacy, continue to impede its widespread adoption [14]. This issue is particularly relevant in rural regions [15], such as Brandenburg in Germany, where long travel distances to specialized care create barriers to timely treatment [16]. Despite the technological potential of TM, a critical question remains: who is both willing and able to integrate TM effectively into clinical practice?

Previous studies suggest that healthcare professionals (HCPs) act as key gatekeepers in the adoption of digital health interventions, with their acceptance or reluctance ultimately determining the extent to which TM is implemented in real-world healthcare settings [17].

Historically, theoretical models such as Rogers' Diffusion of Innovation Theory, the Technology Acceptance Model (TAM), and the Unified Theory of Acceptance and Use of Technology (UTAUT-II) have been employed to explain the adoption of digital health interventions [18]. While these frameworks offer valuable insights, they remain largely descriptive and are limited in their ability to precisely predict TM utilization. With the rise of artificial intelligence (AI), new opportunities are emerging to systematically identify key predictors of TM adoption.

Predictive modeling enables a precise analysis of the factors influencing HCPs' willingness to integrate TM, helping to tailor technologies to clinical needs. Understanding adoption barriers is crucial for developing inclusive cardiology care models and optimizing healthcare structures while mitigating risks associated with suboptimal TM implementation.

Different statistical analysis approaches have been used in the past few decades in order to understand the factors associated with TM use. Most previous investigations of TM adoption among HCPs in cardiology-related fields have employed either qualitative methods [19–21], descriptive statistics [22,23] clustering techniques [24,25], or logistic regression models [26–28]. While these approaches have provided valuable insights, they also carry limitations, particularly in terms of addressing model uncertainty, achieving robust predictive performance, and capturing complex nonlinear relationships between predictors and outcomes.

In this study, we aimed to address these methodological gaps by applying Bayesian model averaging (BMA) alongside modern explainable machine learning (ML) techniques. BMA is designed to identify determinants that are statistically associated with an outcome (e.g., TM use) and provide insights into potential mechanisms and drivers of behavior. In contrast, ML methods such as extreme gradient boosting (XGBoost), prioritize predictive accuracy and are able to capture complex, non-linear relationships that might remain hidden in conventional regression-based approaches. Unlike conventional regression analyses, which generally identify a single "best-fitting" model and disregard the uncertainty surrounding model specification, BMA formally accounts for this uncertainty by averaging across multiple plausible models weighted by their posterior probabilities [29]. This approach yields parameter estimates that are more robust, less sensitive to sampling variability, and therefore more reliable in contexts such as TM adoption, where multiple interdependent factors are at play [17]. In addition, we applied a machine learning approach using XGBoost, which is widely recognized for its strong predictive accuracy, computational efficiency, and capacity to capture nonlinear and high-dimensional dependencies [30–32].

By combining BMA and ML in the present study, we sought to distinguish between explanatory relevance (factors associated with TM use) and predictive relevance (factors that improve classification of TM users vs. non-users). This dual perspective not only provides complementary insights into the determinants of TM adoption among HCPs but also allows us to assess the robustness of findings across distinct analytical frameworks.

## Methods

### Overview

This study is a secondary analysis of previously published data from a web-based survey assessing knowledge, acceptance, and utilization of TM among cardiologists, internists, and general practitioners [33]. The pre-validated 34-item questionnaire covered TM-related topics, including perceived benefits, barriers, and clinical applications. The full list of questionnaire items is provided in Supplementary S1 Table. The primary study identified physician subgroups based on TM attitudes and usage patterns.

## Studied population

Participants were eligible if they met the following inclusion criteria: 1) working in inpatient or outpatient cardiology care, including cardiologists, internists, and general practitioners, 2) practicing in Brandenburg or Berlin, and 3) providing informed consent. Physicians who did not meet these criteria were excluded, along with questionnaires in which less than 50% of the items were completed. Recruitment was conducted through professional email invitations, a QR code published in a regional health journal, and additional outreach via platform X. Data collection took place from May 28, 2021, to February 28, 2022, using the web-based survey tool *LimeSurvey* [34].

## Statistical analysis

All statistical analyses were performed using R software 4.5.0® (R Core Team, Vienna, Austria) for Windows 11©. The identification of TM use determinants followed the same methodology as in a previous work [33].

## Data management

Independent variables were defined based on all columns except the outcome variable TM use. Ordered variables were retained as ordered factors, and nominal categorical variables with more than two levels were one-hot encoded, with one reference category omitted per variable to prevent multicollinearity [24]. Missing values were handled as a separate category, under the assumption that nonresponse may reflect systematic differences in TM use. After these steps, the dataset included 53 independent covariates.

## Bayesian model averaging (BMA) analysis

We implemented a structured three-step pipeline to identify determinants of TM use. First, collinearity screening was conducted by examining pairwise correlations and calculating the variance inflation factor (VIF), with variables exhibiting VIF > 2.5 excluded to mitigate multicollinearity [24]. Next, LASSO regression was applied to reduce dimensionality and retain the most informative variables. Optimal penalization parameters were identified using five-fold cross-validation (CV), and variables with non-zero coefficients at the optimal lambda were retained. These variables were subsequently entered into a BMA framework with logistic regression. BMA averages across all candidate models, weighting by posterior probabilities, thus accounting for model uncertainty (Depaoli et al. 2021). This process provided posterior inclusion probabilities (PIPs) for each feature, indicating the likelihood that the variable has a non-zero effect on TM use. Posterior mean coefficients and standard deviations were calculated across all models (full BMA estimates), and we also summarized results from the single highest posterior probability model and the five highest posterior probability models for comparison. Variables with non-zero PIPs were considered determinants of TM use, with higher PIPs reflecting stronger evidence of association. Visual summaries of group-level differences were produced using radar plots, where continuous and ordinal variables were normalized to a 0–1 range using min–max scaling and summarized by group medians, whereas dichotomous variables were summarized by group proportions. BMA was performed using the BMA package (version 3.18.17) [27]. Regarding priors for BMA, we assumed that all candidate models were equally likely a priori (same prior weight).

## Machine learning analysis

To complement the BMA analysis and leverage the predictive capacity of ML, we applied a binary XGBoost model. Unlike regression models, XGBoost handles multicollinearity through its tree-based structure, so no pre-filtering for collinearity was performed [30–32] To avoid biased evaluation, a nested CV framework was applied. The dataset was first split into a 70% training set and a 30% holdout test set using stratified sampling to preserve the proportion of TM users and non-users. Within the training set, a 10-fold outer CV was performed, with each fold sequentially serving as validation while the remaining folds formed the training subset. Within each outer training fold, a 10-fold inner CV was conducted to

tune hyperparameters. Due to computational constraints, 30 random hyperparameter combinations were tested for each fold, with learning rate ([0.05-1]), maximum depth ([3–7]), minimum sum of instance weights (hessian) needed in a child node ([1–10]), subsample ([0.5-1]), and subsample ratio of columns when constructing each tree ([0.5-1]) varied within pre-specified ranges. The scale_pos_weight parameter was set to the ratio of majority to minority class to account for class imbalance. Hyperparameters were optimized by maximizing the mean area under the receiver operating characteristic curve (AUROC) across inner folds. The best hyperparameter configuration for each outer fold was used to train an XGBoost model, which was then evaluated on the corresponding outer validation set.

Following nested CV, the hyperparameter configuration yielding the highest AUROC across outer folds was selected to train a final XGBoost model on the full training set. This model was evaluated on the independent 30% holdout test set. Evaluation metrics included AUROC with 95% confidence intervals, accuracy with 95% confidence intervals, F1-score, and additional metrics such as precision, recall, and specificity. The F1 score is a measure of predictive performance, combining precision and recall into a single value, ranging from 0 (poor performance) to 1 (perfect performance).

The XGBoost algorithm was computed using the xgboost package version 1.7.8.1 [35]. The caret package version 7.0-1 was used to calculate the confusion matrix [36]. The pROC package version 1.18.5 [37] was used to calculate the AUROC, while the MLmetrics package version 1.1.3 [38] was used to compute evaluation performance metrics not provided by the caret package. The stratified sampling was performed with the splitTools package version 1.0.1 [39]. Model training, cross-validation, and hyperparameter tuning were carried out with the mlexperiments package [40] in combination with the mllrnrs package [41], which provide standardized machine learning pipelines and learner interfaces.

To enhance the interpretability of the XGBoost model, SHapley Additive exPlanations (SHAP) values were calculated using the holdout test set to quantify the contribution of each predictor to the model's output. SHAP values reflect the individual contribution of each predictor by evaluating its marginal impact on the model's output [42,43].Predictors that directed the model toward TM use were considered promoting factors, whereas predictors directing the model toward non-use were considered limiting factors. Directionality was inferred using a hybrid heuristic. For binary and sparse features, mean SHAP values were compared across groups, while for continuous features, generalized additive models (GAMs) were fitted to SHAP values, and the average derivative of the fitted curve was used to determine association direction. When GAMs failed to converge, Spearman's rank correlation between the feature and its SHAP values served as a fallback.

### Sensitivity analysis

A sensitivity analysis was performed by excluding individuals with missing values to evaluate the robustness of our findings.

### Ethics statement

The study was approved by the local ethics committee of the Brandenburg Medical School (E-01–20210304). Data processing was based on the informed consent of the participants in the study. Participation in the study was not remunerated. Personal data were only collected from the participants to be able to process any requests in accordance with current law. These data were deleted after the end of the study. No other personal data were collected [44].

## Results

### Healthcare professional characteristics

A total of 112 healthcare professionals (HCPs) were considered, including 64 (57%) TM users. Compared to our previous work, we included all 112 HCPs to prevent the introduction of a selection bias. There were 54 males (48%), 28 females (25%), 2 non-binaries (1.8%), and 28 persons (25%) who did not declare their gender. Most HCPs were 50–59 years

(29%), followed by HCPs aged 40–49 (21%), 30–39 (14%), ≥ 60 (9%), and 20–29 (4%). There were 63 internal medicine specialists (56%), 47 cardiologists (42%), and 12 general practitioners (11%). On average, HCPs had 800 patients per quarter. For sample characteristics we refer to the recently published study [19].

### BMA analysis

Of the 53 independent covariates initially considered, 32 (60%) were retained after excluding collinear variables. Among these 32, 12 (38%) were selected using Least Absolute Shrinkage and Selection Operator (LASSO) regression and subsequently included in the BMA analysis. Examination of posterior model probabilities revealed model uncertainty. No single model dominated, with the highest posterior model accounting for only 11.0% of the total weight, and the top five models together representing 41.3%. Estimates based on only the top model, or approximated across the top five models, differed from the fully model-averaged estimates (Table 1 and Fig 1). In contrast, posterior means from the full BMA, which incorporate information from the entire model space, provided more stable estimates that were often shrunk toward zero.

A total of 12 determinants of TM use were identified (Table 1 and Fig 2). Positive associations were found for TM knowledge, being a cardiologist, self-identifying as female, perceiving TM as suitable for heart failure and for monitoring an event, considering absence of remuneration and data security as barriers to TM, and perceiving TM as relevant for extending the aftercare interval and for basic ECG monitoring. Negative associations were observed for considering insufficient data on patient benefit as a TM barrier, and for perceiving TM as less suitable in acute events or for extending the aftercare interval (Table 1).

The sensitivity analysis produced similar findings, identifying eight determinants, all of which overlapped with the main analysis and showed consistent directions of association (Figs 2 and S1 and S1 Table). However, in this sensitivity analysis, data security, the relevance of TM for ECG monitoring and aftercare extension, and the unsuitability of TM in acute events were not retained as determinants.

The determinants identified through BMA were used to describe the profiles of HCPs in cardiology who use and do not use TM (Fig 2). Profiles are presented for both the main and sensitivity analyses. The percentages correspond to the proportion of HCPs endorsing each specified response. For example, 100% (the outer line of the radar chart) indicates that all HCPs selected the given response (e.g., self-identifying as female), whereas 0% (the inner line) indicates that none selected it.

**Table 1. Factors identified through BMA and their association with TM use.**

| Determinants | Top model | Top 5 models | BMA | | Posterior inclusion probability (%) | Direction |
|---|---|---|---|---|---|---|
| | mean | mean | mean | sd | | |
| TM knowledge | 1.45 | 1.00 | 1.45 | 0.36 | 100 | Positive association |
| TM barrier: no remuneration possible | 0 | 0 | 0.13 | 0.48 | 8.80 | Positive association |
| TM barrier: data security | 0 | 0 | 0.02 | 0.21 | 2.20 | Positive association |
| TM barrier: insufficient data for the benefit of patients | 0 | 0.13 | -0.27 | 0.72 | 16.5 | Negative association |
| TM use suitable for heart failure | 0 | 0 | 0.17 | 0.46 | 14.1 | Positive association |
| TM use suitable for monitoring an event | 0 | 0.55 | 0.53 | 0.71 | 43.0 | Positive association |
| TM use relevant for basic monitoring of ECG | 0.23 | 0.27 | 0.23 | 0.50 | 21.4 | Positive association |
| TM use relevant for extension of the aftercare interval | 0 | 0 | 0.03 | 0.17 | 3.80 | Positive association |
| TM use less suitable in acute events | 0 | 0.18 | -0.13 | 0.41 | 11.8 | Negative association |
| TM use less suitable for extension of the aftercare interval | 0 | 0 | -0.004 | 0.07 | 0.90 | Negative association |
| Self-identified as female | 1.42 | 1.00 | 1.42 | 0.93 | 80.6 | Positive association |
| Being a cardiologist | 1.26 | 1.00 | 1.26 | 0.77 | 83.5 | Positive association |

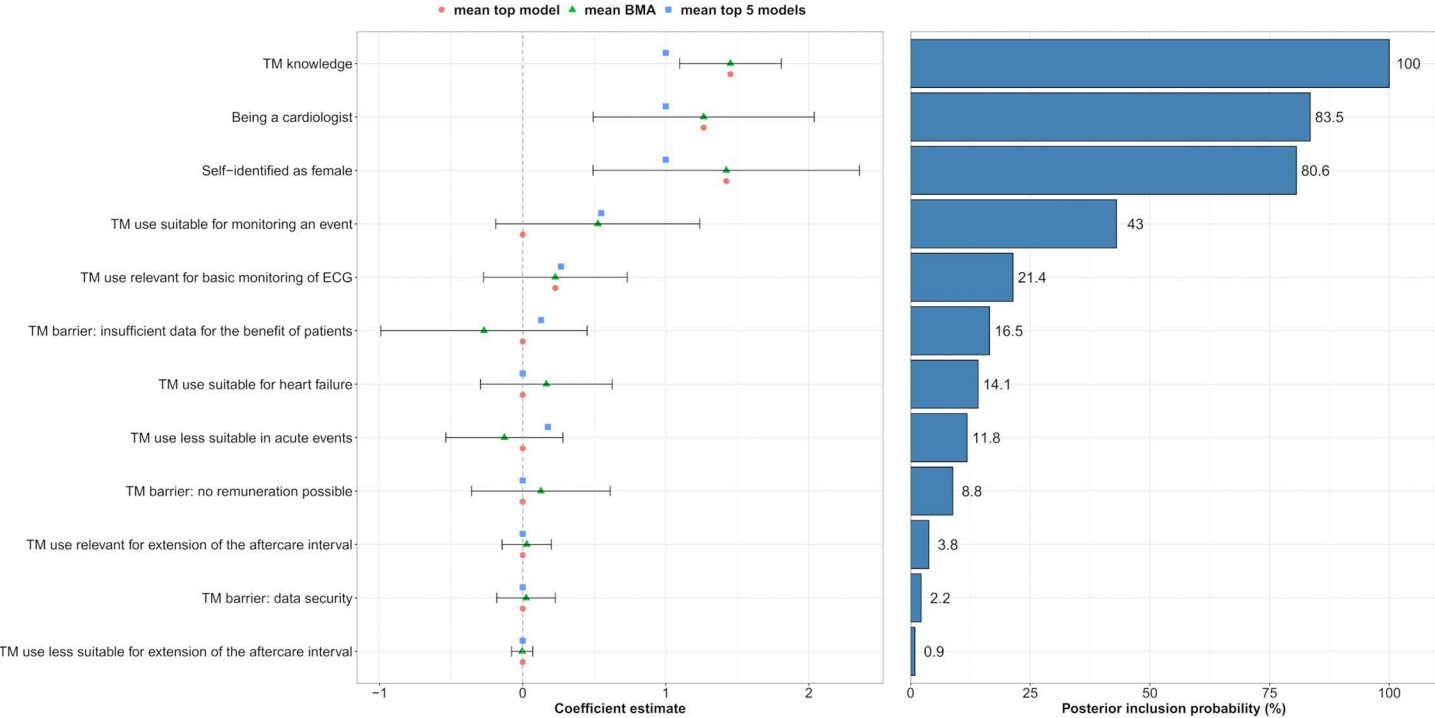

**Fig 1. Coefficient estimates for each predictor variable under three modeling perspectives: the highest posterior probability single model ("Top model"), an approximation averaged over the top five models ("Top 5 models"), and the fully Bayesian model-averaged estimates ("BMA").** The points represent the estimated coefficient effect sizes, while the error bars show the posterior standard deviations derived from the full BMA analysis, reflecting uncertainty in these averaged estimates. The "Top model" and "Top 5 models" estimates do not include uncertainty intervals because these approximations do not capture model-averaged uncertainty. The full "BMA" estimates incorporate all plausible models weighted by their posterior probabilities, shrinking coefficients toward zero when evidence for inclusion is weak and properly reflecting parameter uncertainty.

## Machine learning analysis

The XGBoost model demonstrated strong performance, with an AUROC of 0.88 (95% CI: 0.75–1.00), an accuracy of 0.79 (95% CI: 0.62–0.91), a sensitivity of 79%, a specificity of 80%, and an F1 score of 0.81 (Table 2).

Several of the most influential predictors identified by SHAP analysis (e.g., TM knowledge, being a cardiologist, and female sex) were also among the key determinants highlighted in the BMA analysis (Fig 3 and Table 3). The ML framework identified additional predictors that did not emerge as determinants in the BMA models. For example, perceptions of TM suitability for arrhythmias, pacemaker and defibrillator therapy, and for monitoring blood pressure ranked highly in predictive importance, reflecting the model's ability to capture more complex or non-linear associations. Conversely, factors such as concerns about insufficient patient benefit, which were negatively associated with TM use in BMA, did not contribute substantially to the predictive performance of the ML model.

In terms of directionality, SHAP values indicated that several factors promoted TM use, including TM knowledge, being a cardiologist, female sex, perceiving TM as suitable for arrhythmias and heart failure, higher patient volume per quarter, and viewing TM as relevant for managing complications of drugs and medical devices. In contrast, limiting factors included perceiving TM as unsuitable for acute events or aftercare extension, older age, using TM for own health, and concerns about implementation effort (Table 3 and Fig 3).

The sensitivity analysis yielded broadly consistent findings, although some predictors identified in the main model were not retained, such as perceptions of insufficient remuneration as a barrier or suitability for hypertension (Tables 2, S2 and S3 and S2 Fig). At the same time, new predictors emerged, such as the perception that TM is less suitable for basic ECG

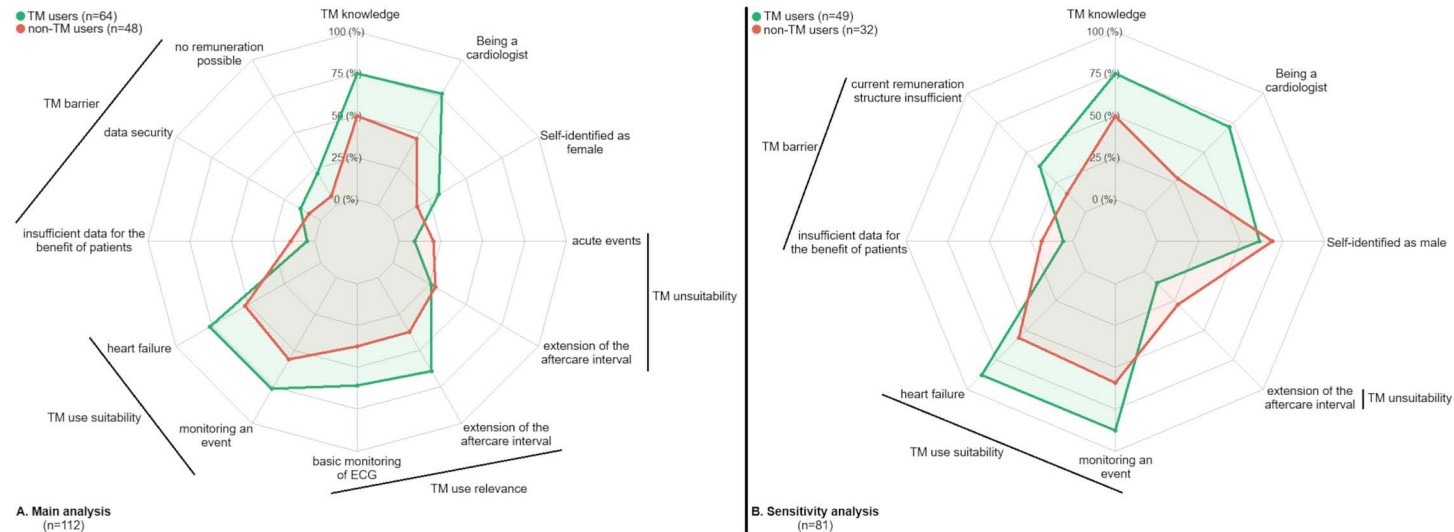

**Fig 2. Profile of healthcare professionals (HCPs) in cardiology using telemedicine (TM) versus non-users.** The radar plot compares key determinants, with TM users (green) and non-users (red).

**Table 2. Performance of the final XGBoost model.**

| Performance metric | Main analysis | Sensitivity analysis |
|---|---|---|
| Accuracy (95% CI) | 0.79 (0.62; 0.91) | 0.84 (0.68; 1.00) |
| AUC (95% CI) | 0.88 (0.75; 1.00) | 0.76 (0.55; 0.90) |
| Kappa | 0.59 | 0.50 |
| No Information Rate (NIR) | 0.56 | 0.60 |
| P value [accuracy > NIR] | 0.004 | 0.07 |
| Mcnemar's test p value | 1.00 | 1.00 |
| Sensitivity/ Recall | 0.79 | 0.80 |
| Specificity | 0.80 | 0.70 |
| Precision/ positive predictive value | 0.83 | 0.80 |
| Negative predictive value | 0.75 | 0.70 |
| F1 score | 0.81 | 0.80 |
| Detection rate | 0.44 | 0.48 |
| Balanced accuracy | 0.79 | 0.75 |

Note: The purpose of the McNemar's test is to test whether the number of false positives and false negatives are statistically equal.

monitoring or for consulting colleagues. Certain predictors shifted directionality. For instance, the number of patients per quarter and willingness to invest in TM showed opposite associations compared with the main model. These discrepancies emphasize that while the overall predictive structure of the ML model was stable, the importance of individual features may vary depending on the population considered.

SHAP value (x-axis) for each HCP and feature are represented with a point. Positive SHAP values imply an impact to the model toward wanting TM use, while negative values impact the model toward not using TM. For each feature, the mean and 95% CI of the absolute SHAP values are reported on the right of the graph. High SHAP value (in absolute value) indicate a

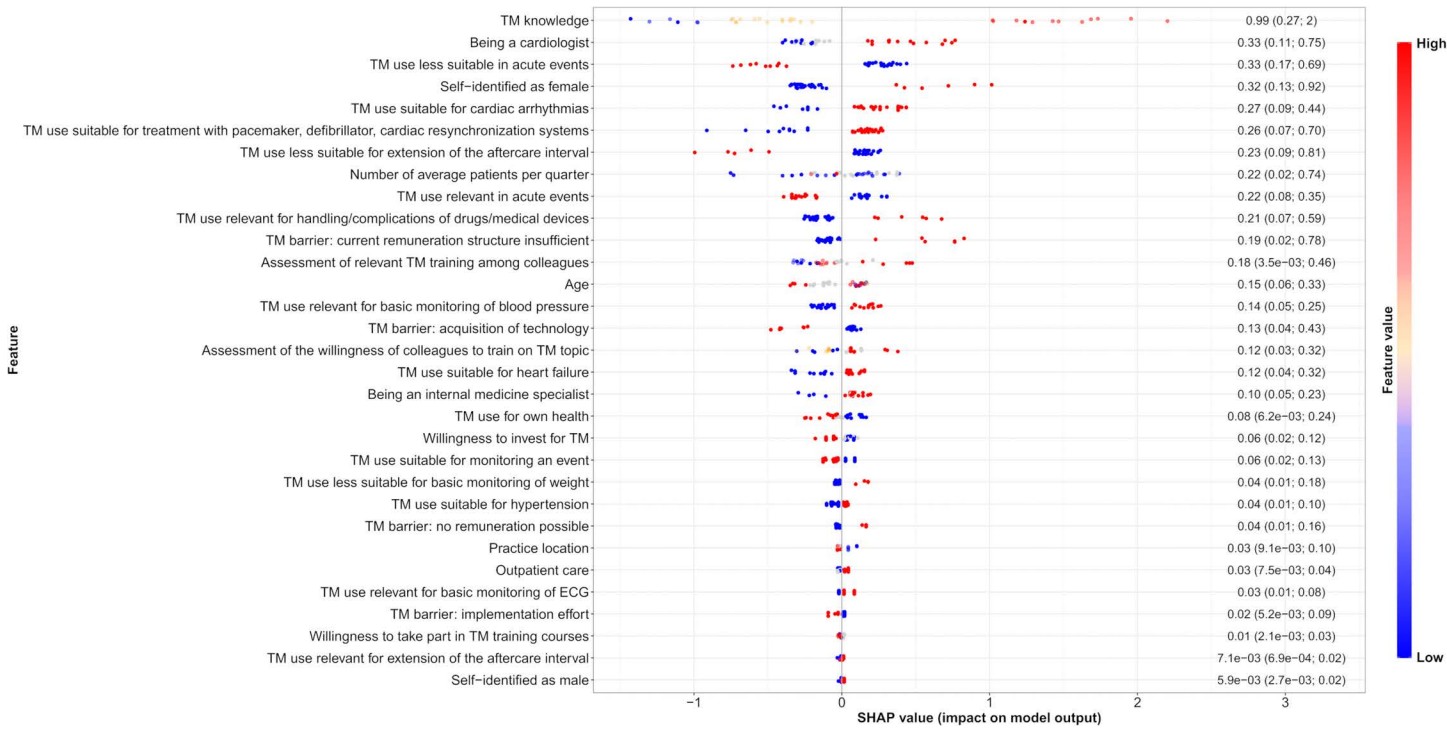

**Fig 3. Feature importance according to SHAP values for the prediction of TM use.**

high impact on the model output. Red colors indicate that an HCP either answered yes to the considered question (y-axis) regarding categorical feature or tend toward high values for continuous features (e.g., age), while blue colors refer either to "no" answers for categorical features or low values for continuous features, and grey colors refer to missing answers.

## BMA and XGBoost comparison

The explanatory and predictive frameworks converged on several key factors while also highlighting different aspects of TM adoption (Table 4). Both BMA and XGBoost identified TM knowledge, female sex, and being a cardiologist as robust determinants of TM use. These variables demonstrated both a consistent statistical association and a strong contribution to predictive performance, suggesting that they represent core drivers of adoption in cardiology practice. Similarly, perceptions that TM is unsuitable for acute events or for extending aftercare intervals were negatively associated with use in BMA and also reduced predictive probability in XGBoost, reinforcing their importance as barriers.

Beyond these points of convergence, the two approaches diverged in meaningful ways. BMA, by design, emphasized parsimonious associations with stable posterior inclusion probabilities, highlighting concerns about insufficient patient benefit as a limiting factor, an effect that did not substantially improve the predictive capacity of the XGBoost model. Conversely, XGBoost identified perceptions of TM suitability for arrhythmias, device monitoring, and blood pressure management as strong predictors, even though these factors did not consistently emerge as explanatory determinants in BMA. These differences underscore the complementary nature of the two methods. BMA provides insights into determinants with stable associations, while ML captures a broader set of variables that enhance classification accuracy, including those with weaker or more context-dependent effects.

Importantly, one feature (perceptions of TM suitability for event monitoring) exhibited incongruent behavior across approaches, showing a positive association in one framework (BMA) but a limiting effect in the other (XGBoost). Such

**Table 3. Top predictors of TM Use identified with XGBoost.**

| Feature | Mean \|SHAP\| (95%CI) | Direction |
| --- | --- | --- |
| TM knowledge | 0.99 (0.27; 2) | Promoting |
| Being a cardiologist | 0.33 (0.11; 0.75) | Promoting |
| TM use less suitable in acute events | 0.33 (0.17; 0.69) | Limiting |
| Self-identified as female | 0.32 (0.13; 0.92) | Promoting |
| TM use suitable for cardiac arrhythmias | 0.27 (0.09; 0.44) | Promoting |
| TM use suitable for treatment with pacemaker, defibrillator, cardiac resynchronization systems | 0.26 (0.07; 0.70) | Promoting |
| TM use less suitable for extension of the aftercare interval | 0.23 (0.09; 0.81) | Limiting |
| Number of average patients per quarter | 0.22 (0.02; 0.74) | Promoting |
| TM use relevant in acute events | 0.22 (0.08; 0.35) | Limiting |
| TM use relevant for handling/complications of drugs/medical devices | 0.21 (0.07; 0.59) | Promoting |
| TM barrier: current remuneration structure insufficient | 0.19 (0.02; 0.78) | Promoting |
| Assessment of relevant TM training among colleagues | 0.18 (3.5e-03; 0.46) | Promoting |
| Age | 0.15 (0.06; 0.33) | Limiting |
| TM use relevant for basic monitoring of blood pressure | 0.14 (0.05; 0.25) | Promoting |
| TM barrier: acquisition of technology | 0.13 (0.04; 0.43) | Limiting |
| Assessment of the willingness of colleagues to train on TM topic | 0.12 (0.03; 0.32) | Promoting |
| TM use suitable for heart failure | 0.12 (0.04; 0.32) | Promoting |
| Being an internal medicine specialist | 0.10 (0.05; 0.23) | Promoting |
| TM use for own health | 0.08 (6.2e-03; 0.24) | Limiting |
| TM use suitable for monitoring an event | 0.06 (0.02; 0.13) | Limiting |
| Willingness to invest for TM | 0.06 (0.02; 0.12) | Limiting |

discrepancies highlight the value of applying both explanatory and predictive methodologies, as they draw attention to variables whose influence on TM adoption may depend on context, measurement, or modeling assumptions.

## Discussion

### Main findings

This study applied BMA and XGBoost to move beyond confirming general associations by estimating the relative importance of predictors, producing more robust effect estimates, and exploring potential non-linear relationships to provide a more nuanced picture of TM adoption among HCPs. The findings can be summarized as follows. First, across both frameworks, TM knowledge, female sex, and being a cardiologist consistently emerged as core drivers of TM use. Second, the lack of remuneration was identified as a main barrier for TM use in both models. Third, the two approaches also highlighted distinct aspects of TM adoption. BMA emphasized concerns about insufficient patient benefit as a limiting factor, whereas XGBoost identified perceptions of TM suitability for arrhythmias, device monitoring, and blood pressure management as strong predictors. This complementarity underscores the value of combining explanatory and predictive perspectives, explanatory models reveal stable associations that clarify underlying mechanisms, while predictive models capture additional and sometimes non-linear factors that improve classification.

### Comparison with previous work

Most previous investigations of TM adoption among HCPs in cardiology-related fields have relied on qualitative methods [19–21], descriptive statistics [22,23], clustering techniques, or classical logistic regression models [26–28]. While

**Table 4. Comparison of the feature impact on TM use depending on the statistical method use.**

| Domain | Feature | BMA (association) | XGBoost (prediction) |
|---|---|---|---|
| Personal | TM knowledge | promoting | promoting |
| Personal | Self-identified as female | promoting | promoting |
| Personal | Age | | limiting |
| Personal | TM use for own health | | limiting |
| Personal | Willingness to invest for TM | | limiting |
| Occupational | Being a cardiologist | promoting | promoting |
| Occupational | Being an internal medicine specialist | | promoting |
| Occupational | Number of average patients per quarter | | promoting |
| Occupational | Assessment of the willingness of colleagues to train on TM topic | | promoting |
| Occupational | Assessment of relevant TM training among colleagues | | promoting |
| TM use unsuitability | Acute events | limiting | limiting |
| TM use unsuitability | Extension of the aftercare interval | limiting | limiting |
| TM use suitability | Cardiac arrhythmias | | promoting |
| TM use suitability | Monitoring an event | promoting | limiting |
| TM use suitability | Heart failure | promoting | promoting |
| TM use suitability | Treatment with pacemaker, defibrillator, cardiac resynchronization systems | | promoting |
| TM use relevance | Acute events | | limiting |
| TM use relevance | Handling/complications of drugs/medical devices | | promoting |
| TM use relevance | Basic monitoring of blood pressure | | promoting |
| TM use relevance | Basic monitoring of ECG | promoting | |
| TM use relevance | Extension of the aftercare interval | promoting | |
| TM barrier | Acquisition of technology | | limiting |
| TM barrier | No remuneration possible | promoting | promoting |
| TM barrier | Data security | promoting | |
| TM barrier | Insufficient data for the benefit of patients | limiting | |
| TM barrier | Current remuneration structure insufficient | | promoting |

**Note:** empty cells refer to features that were not selected in the models.

these approaches provided valuable insights into general associations, they are limited in addressing model uncertainty, assessing the relative importance of predictors, and capturing complex non-linear relationships. More recently, advanced modelling approaches have been applied to cardiology-related telemedicine data. Koos et al. (2022) used hierarchical logistic regression with cross-validation to disentangle the contributions of patient-, clinician-, and visit-level characteristics to telemedicine use in a large academic cardiovascular center, showing that the individual clinician seen was by far the strongest predictor (AUC = 0.83) [45]. In contrast, Tisdale et al. (2024) analyzed more than 220,000 patients in the U.S. Veterans Health Administration using multilevel logistic regression and found that 40.5% of the variability in telehealth use was attributable to the patient level, 30.8% to the clinician level, and 7% to the facility level [46]. These studies highlight the hierarchical structure of TM adoption but modelled mainly linear effects and did not quantify the relative contribution of individual predictors or explore potential non-linear patterns. Although our survey included only HCPs and could therefore capture only clinician-centered domains, both models consistently identified TM knowledge as the most important factor for TM use. This aligns with prior evidence showing that clinicians' self-reported technological proficiency, comfort with TM platforms, and positive attitudes toward TM are among the strongest predictors of both current and future TM use, independent of demographic or professional background [47–51].

To the best of our knowledge, no prior work has applied approaches capable of assessing the relative importance of predictors and non-linear relationships to provide a more nuanced understanding of TM adoption dynamics among health-care professionals in cardiology. Nevertheless, our findings can be robustly situated within the existing literature, as they align with consistently reported determinants of TM use.

Both models in our analysis consistently identified heart failure as a condition perceived to be particularly suitable for TM. This aligns with evidence from clinical trials such as TIM-HF II, which demonstrated the effectiveness of structured telemedical care in heart failure management [45,46]. It is plausible that our respondents considered TM most appropriate in scenarios they were already familiar with, reflecting existing clinical experience rather than openness to entirely new use cases. Interestingly our BMA framework highlighted concerns about insufficient patient benefit as a limiting factor, this likely reflects uncertainty or limited awareness rather than an actual lack of evidence. TM in cardiology has demonstrated significant clinical benefits. A systematic review by Mohammadzadeh et al. found that telecardiology interventions facilitate early diagnosis and treatment, ultimately reducing mortality in patients with cardiovascular disease. Additionally, TM lowers healthcare costs and improves patient quality of life and satisfaction [51].

Similarly, studies by Raes et al. and Farabi et al. confirmed that telemonitoring for patients with cardiac implantable electronic devices (CIEDs), as well as telemedicine integrated with usual care, improves clinical outcomes while reducing healthcare costs [52,53]. Robust evidence further shows that telemedicine enhances clinical outcomes, patient satisfaction, and healthcare utilization in cardiology, particularly in chronic conditions such as heart failure, hypertension, and arrhythmias [14,52,53].

One possible explanation for the mixed perceptions of telemedicine in acute care contexts is that clinicians may differentiate between specific types of acute scenarios. Telemedicine may be perceived as appropriate for certain situations, such as diagnostic consultation or initial triage, while being considered less suitable for scenarios requiring immediate physical intervention. As the questionnaire did not explicitly distinguish between different subtypes of acute cardiovascular situations, our findings reflect general perceptions rather than evaluations of specific clinical contexts. Future research should therefore investigate in greater detail how clinicians assess the suitability of telemedicine across different acute care scenarios.

Taken together, these findings suggest that the negative association observed in our model may reflect knowledge gaps or misconceptions rather than a lack of supporting evidence, underscoring the need for targeted education and awareness initiatives to strengthen confidence in telemedical approaches among healthcare professionals.

Both of our models identified lack of remuneration as the strongest barrier to TM use. This aligns with prior studies showing that unclear or insufficient reimbursement is a key obstacle to TM adoption among clinicians in Germany. Gehrmann et al. (2025) reported that more than half of cardiologists, internists, and general practitioners cited inadequate reimbursement as a major barrier [53] and similar findings were reported by Muehlensiepen et al. (2021) in Rheumatologist [54] and Brauns and Loos (2015) [55]. Despite recent reforms, reimbursement for telemedical services remains fragmented and limited [56] and physicians continue to call for remuneration models equivalent to in-person care [57]. Our findings thus reinforce the view that structural financial barriers remain a major bottleneck for sustainable TM implementation for German HCPs.

## Strengths and limitations

A key strength of our study is the application of established sophisticated statistical methodologies to real-world survey data collected from actual healthcare professionals. Those real-world data allow us to identify insights grounded in actual clinical perspectives of HCP. Given the complexity of the data and the many potential predictor variables, uncertainty about the best predictive model is inevitable.

Taken together, the combination of BMA and ML enhances both the inferential rigor and predictive capacity of our analysis. Whereas prior studies have largely confirmed general associations between HCP characteristics and TM adoption,

our study adds value by providing stronger evidence regarding the relative importance of predictors, generating more reliable effect estimates, and uncovering nonlinear relationships that yield a more nuanced understanding of adoption dynamics. By moving beyond descriptive or exploratory approaches, our work contributes a more robust and actionable analytical framework for advancing clinical and policy strategies in TM.

In our BMA analysis, the five models with the highest posterior probabilities together accounted for only 41.3% of the total model weight, indicating considerable model uncertainty. Many variables displayed larger effect sizes in the "top model" or across the "top five models" compared with the full BMA estimates, illustrating how reliance on a small subset of models can exaggerate associations by neglecting the broader model space. Because the posterior probabilities of the leading models were relatively low, coefficients derived from them, essentially maximum likelihood estimates conditional on a single specification, risk overstating evidence. In contrast, the BMA posterior means integrate information across the entire space of plausible models, shrinking coefficients toward zero when support for a predictor is weak and thus yielding more stable and reliable estimates. These findings underscore that restricting inference to a single "best" model, or even to a handful of high-probability models, would disregard most of the available information. By averaging over the complete model space, BMA provides more robust inference and guards against spurious conclusions driven by arbitrary model selection.

We further explore variable relevance by additionally applying an explainable ML approach, combining XGBoost, a powerful ML algorithm widely applied in predictive modeling tasks [58,59] with SHAP to enhance interpretability. The use of nested cross-validation represents a methodological strength, as it separates hyperparameter tuning from model evaluation and thereby reduces the risk of overfitting. Nevertheless, despite this safeguard, the relatively small sample size (n = 112) and the high number of initial variables (n = 53) could still allow the model to capture dataset-specific patterns, potentially inflating variable importance. The model's strong predictive performance highlights the relevance of the identified factors and suggests that targeted interventions, such as structured training and administrative reforms, could effectively address adoption barriers and support broader TM integration in clinical practice. Although we did not validate our model using an independent dataset or multiple resampling iterations, the nested CV framework provides a reliable estimate of predictive performance and helps ensure that the identified predictors are supported within the training data. Our dataset includes a heterogeneous group of HCPs from diverse clinical settings, supporting the relevance of the findings. However, generalizability remains limited by the relatively small sample size, geographic restriction to two regions in Germany, and absence of external validation. Future studies should aim to replicate and extend this predictive framework using larger, independent datasets to assess its applicability across broader healthcare contexts.

A key limitation of our study is the cross-sectional nature of the survey data, which limits causal interpretation. The aim of this study was not to establish predictive causality, but rather to explore and characterize distinct profiles of TM users. Some of the variables included as predictors, such as TM knowledge, perceptions of TM suitability for heart failure or arrhythmias, experience with cardiac devices, and considerations of remuneration, reflect current practices and attitudes rather than factors that necessarily preceded TM adoption. As such, the predictive models in this study estimate associations with existing TM use, rather than truly forecasting future adoption. A longitudinal design, in which TM use is measured prospectively, would allow for a more rigorous evaluation of causal relationships and the development of predictive models that identify determinants prior to adoption. Future research should validate our findings in independent and temporally structured datasets, ideally with repeated measures of TM use and predictors over time. Such studies would enable more accurate prediction of TM adoption and help distinguish factors that actively drive uptake from those that are concomitant with current usage.

Because both BMA and XGBoost involve stochastic processes, the specific random seed used may influence the results. As a consequence, using a different seed could yield slightly different model estimates, variable importance rankings, or performance metrics, introducing potential variability in the findings.

A potential limitation of our web-based survey is the likelihood of a positive selection bias, as physicians already interested in digitalization, telemedicine, or telecardiology may have been more inclined to participate. While recruitment

included digital invitations, an additional outreach via the print magazine *KV-intern* (Kassenärztliche Vereinigung Brandenburg) aimed to reach all physicians registered with the Medical Association Brandenburg, mitigating but not eliminating this bias.

### Implications

Our dual statistical perspective strengthens both the interpretability and practical utility of the findings, providing insights into why TM is adopted as well as who is most likely to adopt it.

By revealing which factors most strongly shape TM adoption among HCPs, our models can help target interventions more effectively. Based on our findings structured training to increase TM knowledge should be prioritized.

Notably, lack of remuneration emerged as a strong barrier in both models, underscoring the need to develop reimbursement frameworks that directly address this structural obstacle. This aligns with position statements from the American College of Cardiology, the European Heart Rhythm Association, and the American Heart Association, which emphasize that resolving reimbursement and regulatory issues is critical for sustainable TM adoption [3]. More broadly, our approach offers a scalable analytical framework for health system planners and policymakers to identify leverage points for promoting telemedicine uptake in other specialties and settings.

### Conclusion

This study applied both Bayesian model averaging and machine learning to identify and predict key drivers of telemedicine use among healthcare professionals in cardiology. Across both analytical frameworks, telemedicine knowledge, being a cardiologist, and female gender emerged as consistent and robust predictors of use. While BMA emphasized stable statistical associations, such as concerns about insufficient patient benefit, machine learning captured additional nonlinear and context-sensitive predictors, including perceptions of telemedicine suitability for specific cardiac conditions like arrhythmias and device monitoring. Importantly, both models confirmed that concerns around insufficient remuneration remain a major barrier to adoption. Together, these findings offer a more nuanced and data-driven understanding of telemedicine adoption dynamics, informing future implementation strategies. By highlighting both explanatory determinants and predictive indicators, this study supports the development of targeted interventions, such as structured training and reimbursement reform, to promote more equitable and sustainable integration of telemedicine into cardiology practice.

### Supporting information

**S1 Table. Full list of questionnaire items.** Complete list of all 34 items included in the pre-validated web-based survey assessing knowledge, acceptance, and utilization of telemedicine among healthcare professionals in cardiology care.
(DOCX)

**S2 Table. BMA results from sensitivity analysis.** Determinants of telemedicine use identified through Bayesian model averaging after exclusion of participants with missing values, including posterior inclusion probabilities and coefficient estimates.
(DOCX)

**S3 Table. XGBoost predictor importance from sensitivity analysis.** Top predictors of telemedicine use identified by the XGBoost model in the sensitivity analysis, including mean absolute SHAP values and directionality.
(DOCX)

**S1 Fig. BMA radar plot from sensitivity analysis.** Profile comparison of telemedicine users versus non-users based on determinants identified through Bayesian model averaging after exclusion of participants with missing values.
(TIF)

**S2 Fig. SHAP feature importance plot from sensitivity analysis.** Feature importance according to SHAP values for the prediction of telemedicine use in the sensitivity analysis excluding participants with missing values.
(TIF)

**S1 Data. Anonymized Survey Data Set.** De-identified individual-level response data from the web-based survey among healthcare professionals in cardiology care.
(CSV)

## Author contributions

**Conceptualization:** Pascal Petit, Jonathan Nübel, Anja Haase Fielitz, Nicolas Vuillerme, Felix Muehlensiepen.

**Data curation:** Pascal Petit.

**Formal analysis:** Pascal Petit.

**Funding acquisition:** Nicolas Vuillerme, Felix Muehlensiepen.

**Investigation:** Jonathan Nübel, Josephine Walter, Anja Haase-Fielitz, Felix Muehlensiepen.

**Methodology:** Pascal Petit, Jonathan Nübel, Nicolas Vuillerme, Felix Muehlensiepen.

**Project administration:** Felix Muehlensiepen.

**Validation:** Pascal Petit.

**Visualization:** Pascal Petit.

**Supervision:** Nicolas Vuillerme, Felix Muehlensiepen.

**Writing – original draft:** Pascal Petit, Jonathan Nübel, Josephine Walter, Christian Butter, Martin Heinze, Yuriy Ignatyev, Nicolas Vuillerme, Felix Muehlensiepen.

**Writing – review & editing:** Pascal Petit, Jonathan Nübel, Felix Muehlensiepen.

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
