## [Decision Letter · Decision Letter 0]

26 Aug 2025

Response to Reviewers'. This file does not need to include responses to any formatting updates and technical items listed in the 'Journal Requirements' section below.'. This file does not need to include responses to any formatting updates and technical items listed in the 'Journal Requirements' section below.* A marked-up copy of your manuscript that highlights changes made to the original version. You should upload this as a separate file labeled 'Revised Manuscript with Track Changes'.'.* An unmarked version of your revised paper without tracked changes. You should upload this as a separate file labeled 'Manuscript'.'. If you would like to make changes to your financial disclosure, competing interests statement, or data availability statement, please make these updates within the submission form at the time of resubmission. Guidelines for resubmitting your figure files are available below the reviewer comments at the end of this letter. We look forward to receiving your revised manuscript. Kind regards, Onicio Batista Leal-NetoAcademic EditorPLOS Digital Health Onicio Leal-NetoAcademic EditorPLOS Digital Health Leo Anthony CeliEditor-in-ChiefPLOS Digital Healthorcid.org/0000-0001-6712-6626 **Journal Requirements:** If the reviewer comments include a recommendation to cite specific previously published works, please review and evaluate these publications to determine whether they are relevant and should be cited. There is no requirement to cite these works unless the editor has indicated otherwise.  **Additional Editor Comments (if provided):****Reviewers' Comments:** Reviewer's Responses to Questions

**Comments to the Author**

1. Does this manuscript meet PLOS Digital Health’s publication criteria? Is the manuscript technically sound, and do the data support the conclusions? The manuscript must describe methodologically and ethically rigorous research with conclusions that are appropriately drawn based on the data presented.? Is the manuscript technically sound, and do the data support the conclusions? The manuscript must describe methodologically and ethically rigorous research with conclusions that are appropriately drawn based on the data presented.

Reviewer #1: No

Reviewer #2: Yes

Reviewer #3: Yes

2. Has the statistical analysis been performed appropriately and rigorously?

Reviewer #1: I don't know

Reviewer #2: No

Reviewer #3: Yes

3. Have the authors made all data underlying the findings in their manuscript fully available (please refer to the Data Availability Statement at the start of the manuscript PDF file)?

The PLOS Data policy requires authors to make all data underlying the findings described in their manuscript fully available without restriction, with rare exception. The data should be provided as part of the manuscript or its supporting information, or deposited to a public repository. For example, in addition to summary statistics, the data points behind means, medians and variance measures should be available. If there are restrictions on publicly sharing data—e.g. participant privacy or use of data from a third party—those must be specified.requires authors to make all data underlying the findings described in their manuscript fully available without restriction, with rare exception. The data should be provided as part of the manuscript or its supporting information, or deposited to a public repository. For example, in addition to summary statistics, the data points behind means, medians and variance measures should be available. If there are restrictions on publicly sharing data—e.g. participant privacy or use of data from a third party—those must be specified.

Reviewer #1: No

Reviewer #2: No

Reviewer #3: No

4. Is the manuscript presented in an intelligible fashion and written in standard English?

Reviewer #1: Yes

Reviewer #2: Yes

Reviewer #3: Yes

Reviewer #1: The authors incorporated statistical and machine learning approaches to investigate the factors associated with the use of telemedicine using the survey data from 112 healthcare professionals. Univariate analysis was first employed to select the factor most strongly associated with the TM use. This is followed by multivariate analyses, including Bayesian Model Averaging, and Shapley value analysis using the XGBoost. While this study raises an interesting question, its utility, and its additional benefit it brings to the clinical community compared to existing studies, remain unclear. In addition, the methodologies in this manuscript were not described in sufficient details, thus making judging the soundness of their methodologies difficult in its current form. Overall, this manuscript requires substantial improvement before it can be considered for publication.

Major concerns:

1) Essentially, this study employed standard statistical and machine learning approaches to analyze the survey data collected from a previous study from the same group, and arrived at a very similar conclusion. Although the authors have provided extensive discussions, the additional benefits this study brings compared to existing studies still need to be further clarified and strengthened. For example, what are the strengths of the methodologies in this study compared to those from the previous studies? Or are there any novel insights that this study provides that were not found by the analytical approaches used in previous studies?

2) Overall, the methodology and the analyses in this study need to be described in much more specific details in order to properly assess the validity of this study, as various crucial information seems to be missing, and some of the descriptions are also unclear and confusing. My specific concerns about these descriptions will be detailed in the following comments, but here are some of the most relevant concerns:

a) To properly evaluate a machine learning model, the cohort used for reporting the performance of the predictive models should never be used during the variable selection and the model training. While the author used cross-validation in the training process, it is unclear whether the variable selection via univariate analysis was conducted using the entire cohort, or within the training cohort of each fold. If it is the former case, the testing partition would have been inadvertently used during the variable selection stage, which would result in information leakage and thereby inflate the performance metrics. Please clarify the independence of data partition in each stage of the analysis. If there is potential information leakage, the authors could either conduct variable selection only within the training data of each fold, or use an independent evaluation cohort for assessing model performance.

b) The 34 items of the questionnaire, which is the central part of the study design, should be provided in the manuscript. In addition to providing the complete set of questionnaire items, how the items related to TM barriers were defined requires further clarification. According to the survey questions in reference 19, the items related to the TM barriers are under the question “What prevents you from using telemedicine? (Multiple selections were possible.)“ Does answering this question directly indicate that the participant did not use telemedicine? If so, the link between TM use and these barrier-related variables should be self-evident. In addition, there are some barrier-related variables that somehow “promote” the use of TM for these participants (according to Table 4). If these barrier-related variables reflect the participating clinicians’ personal use of TM rather than their opinion on the TM use as a public agenda (as indicated by reference 19), this finding will be very counterintuitive and contradicts the authors’ own conclusions in the Discussion section that “TM remains underutilized due to inadequate reimbursement.”

Specific comments:

1) In the Methods section, Variable Selection subsection:

a) “One-hot encoding was applied to all categorical variables with more than two categories.” Judging from the results presented by the authors, ordinal variables were also treated as categorical variables, which limited the explainability of the results. For example, only the age group between 40 and 49 years was considered relevant to predicting TM use, whereas how younger and older age groups are associated with TM use remained unknown. Using approaches that properly handle ordinal variables would greatly improve the explainability of their findings.

b) As mentioned in the Major Concerns, the cohort used for reporting the final model performance (Table 2) should not be used during variable selection and the model training. Please clarify if this is the case in the manuscript.

2) In the Methods section, Machine learning analysis:

a) On “A balanced 70%/30% train/test split was created using downsampling, ensuring no HCPs appeared in both training and test datasets.”: Please be more specific about how the downsampling was performed.

i) By creating a “balanced” training/testing split, did the author intend to convey that the number of HCPs who use TM equals those who do not use TM?

ii) Was the downsampling conducted completely at random? In addition, was the random downsampling performed independently for each fold of the cross-validation? Or done only once prior to cross-validation?

iii) Was the testing set also downsampled? If so, it may inadvertently introduce unwanted selection bias, and the reported values for positive/negative will no longer be valid since it alters the data distribution

iv) “ensuring no HCPs appeared in both training and test datasets” seems to suggest that there are cases where an HCP has filled multiple questionnaires. Please specify if this is true and justify the use of multiple questionnaires per HCP if needed.

b) It was suggested in the abstract that nested cross-validation was used. However, how such nested cross-validation was conducted was either not described in sufficient detail or was described inaccurately. A truly nested cross-validation would consist of an outer loop and an inner loop of the cross-validation. Nevertheless, there is no clear description of how such nested cross-validation was conducted. Additionally, regarding “Final tuning hyperparameter values were chosen based on the best out-of-fold AUROC. Model performance was assessed using AUROC and F1 scores.” Please confirm if the data partition used for selecting the hyperparameter is different from the partition used for assessing the final model performance.

c) Since there are multiple XGBoost models from the 10-fold cross-validation, please clarify how the Shapley values were combined and presented.

3) In the Results section:

a) The purpose of each analysis was not clearly stated in the main text, and the writing should be improved to provide a coherent flow of logic.

b) On the subsection title, “Patient characteristics”: the participants in this study are not patients but healthcare professionals instead.

c) Methods such as BMA and XGBoost should be cited upon their first occurrence.

d) Table 2: The purpose of McNemar's test was not described in the manuscript.

e) Some of the findings also seem to be self-contradictory. Most notably, both “TM use relevant in acute events” and “TM use less suitable in acute events” were considered as limiting factors in Table 3.

f) Table 4 shows a marked divergence between the findings of the two analytical approaches. Although the authors partially addressed this by focusing solely on the congruent findings across these two approaches, it would be more methodologically sound to conduct dedicated sensitivity analyses to investigate how much the findings vary between different initialization criteria or different folds of the cross-validation.

4) In the Discussion section, Strengths and Limitations subsection:

a) “advanced statistical methodologies” is an overstatement, since BMA has been used for decades.

b) “The five models with the highest posterior probabilities (CPPs) together accounted for only 24.1% of the total model weight…This reinforces the importance of our BMA approach, which averages across the full set of plausible models, thereby providing effect estimates that are more robust and less sensitive to arbitrary variable inclusion decisions.” A low CPP when using the top-five models does not automatically lead to the conclusion that BMA can provide more robust estimates when using the full set of plausible models. Quantitative evidence needs to be provided.

Reviewer #2: Petit and colleagues submitted the manuscript : "Predicting telemedicine use among healthcare professionals in cardiology: a Bayesian model averaging and machine learning analysis of German survey data". Usage of telemedicine in modern medicine is a relevant topic. This manuscript is nicely written in logical order and easy to follow. It analyzes variables that influence implementation of telemedicine in Germany.

However, this title doesn't fully match the text of the manuscript. The authors mentioned machine learning (ML) models such as XGBoost, but the results of the ML testing were not fully analyzed nor explained in details in the manuscript. ML model was not meaningfully incorporated in this study, it's not clear why did authors perform this ML analysis. This study could be completely explained by traditional statistical models. Bayes method is one of them, but significance of these findings is not well explained. If the authors want to keep ML analysis in the text, they need to better elaborate the findings and to compare results with at least 1-2 similar studies where ML models were used.

It's not easy to recognize contribution of this manuscript. Most of findings are well known.

I would suggest the authors to complete ML part with better explanations especially in the discussion section. And comparison of ML findings with other similar studies. Or the alternative is to completely remove the ML study from the text and expand statistical analysis and provide significance of the findings using adequate statistical methods. After correction of the text as suggested they could resubmit the manuscript.

Reviewer #3: The paper presents a secondary analysis of survey data from German healthcare professionals to determine the factors that influence and forecast the use of telemedicine in cardiology. The work is technically sound and provides insightful information about the uptake of digital health from a data science standpoint. Below are my suggestions to the authors to improve the paper:

1. To meet PLOS's data sharing guidelines, think about storing anonymised survey data in a public repository.

2. The rigor section would be strengthened by a brief explanation of possible overfitting hazards despite nested cross-validation, given the relatively small sample size (n=112) and large number of initial variables (n=53).

3. Although the authors point out that there isn't an independent test set, the significance of the paper would be increased with a clear plan for future validation.

4. While handling missing data as distinct categories makes sense, a sensitivity analysis could bolster robustness assertions even further.

5. Although figures are instructive, interpretability might be improved by including confidence intervals for SHAP value ranks or by including an additional plot that illustrates feature interaction effects.

**Do you want your identity to be public for this peer review?** For information about this choice, including consent withdrawal, please see our Privacy Policy..

Reviewer #1: **Yes:** Shih-Yen LinShih-Yen LinShih-Yen LinShih-Yen Lin

Reviewer #2: No

Reviewer #3: **Yes:** Mohammad Ali TakallouMohammad Ali TakallouMohammad Ali TakallouMohammad Ali Takallou

**Figure resubmission:** While revising your submission, please upload your figure files to the Preflight Analysis and Conversion Engine (PACE) digital diagnostic tool, https://pacev2.apexcovantage.com/. PACE helps ensure that figures meet PLOS requirements. To use PACE, you must first register as a user. Registration is free. Then, login and navigate to the UPLOAD tab, where you will find detailed instructions on how to use the tool. If you encounter any issues or have any questions when using PACE, please email PLOS at figures@plos.org. Please note that Supporting Information files do not need this step. If there are other versions of figure files still present in your submission file inventory at resubmission, please replace them with the PACE-processed versions. **Reproducibility:** To enhance the reproducibility of your results, we recommend that authors of applicable studies deposit laboratory protocols in protocols.io, where a protocol can be assigned its own identifier (DOI) such that it can be cited independently in the future. Additionally, PLOS ONE offers an option to publish peer-reviewed clinical study protocols. Read more information on sharing protocols at https://plos.org/protocols?utm_medium=editorial-email&utm_source=authorletters&utm_campaign=protocols To enhance the reproducibility of your results, we recommend that authors of applicable studies deposit laboratory protocols in protocols.io, where a protocol can be assigned its own identifier (DOI) such that it can be cited independently in the future. Additionally, PLOS ONE offers an option to publish peer-reviewed clinical study protocols. Read more information on sharing protocols at https://plos.org/protocols?utm_medium=editorial-email&utm_source=authorletters&utm_campaign=protocols

---

## [Decision Letter · Decision Letter 1]

10 Mar 2026

Response to Reviewers'. This file does not need to include responses to any formatting updates and technical items listed in the 'Journal Requirements' section below.'. This file does not need to include responses to any formatting updates and technical items listed in the 'Journal Requirements' section below.* A marked-up copy of your manuscript that highlights changes made to the original version. You should upload this as a separate file labeled 'Revised Manuscript with Track Changes'.'.* An unmarked version of your revised paper without tracked changes. You should upload this as a separate file labeled 'Manuscript'.'. If you would like to make changes to your financial disclosure, competing interests statement, or data availability statement, please make these updates within the submission form at the time of resubmission. Guidelines for resubmitting your figure files are available below the reviewer comments at the end of this letter. We look forward to receiving your revised manuscript. Kind regards, Mathew V. Kiang, PhDSection EditorPLOS Digital Health Mathew KiangSection EditorPLOS Digital Health Leo Anthony CeliEditor-in-ChiefPLOS Digital Healthorcid.org/0000-0001-6712-6626  **Journal Requirements:** If the reviewer comments include a recommendation to cite specific previously published works, please review and evaluate these publications to determine whether they are relevant and should be cited. There is no requirement to cite these works unless the editor has indicated otherwise.  **Additional Editor Comments (if provided):****Reviewers' Comments:** Reviewer's Responses to Questions

**Comments to the Author**

Reviewer #1: (No Response)

Reviewer #3: All comments have been addressed

publication criteria? Is the manuscript technically sound, and do the data support the conclusions? The manuscript must describe methodologically and ethically rigorous research with conclusions that are appropriately drawn based on the data presented.? Is the manuscript technically sound, and do the data support the conclusions? The manuscript must describe methodologically and ethically rigorous research with conclusions that are appropriately drawn based on the data presented.

Reviewer #1: Yes

Reviewer #3: Yes

3. Has the statistical analysis been performed appropriately and rigorously?

Reviewer #1: Yes

Reviewer #3: Yes

4. Have the authors made all data underlying the findings in their manuscript fully available (please refer to the Data Availability Statement at the start of the manuscript PDF file)?

The PLOS Data policy requires authors to make all data underlying the findings described in their manuscript fully available without restriction, with rare exception. The data should be provided as part of the manuscript or its supporting information, or deposited to a public repository. For example, in addition to summary statistics, the data points behind means, medians and variance measures should be available. If there are restrictions on publicly sharing data—e.g. participant privacy or use of data from a third party—those must be specified.requires authors to make all data underlying the findings described in their manuscript fully available without restriction, with rare exception. The data should be provided as part of the manuscript or its supporting information, or deposited to a public repository. For example, in addition to summary statistics, the data points behind means, medians and variance measures should be available. If there are restrictions on publicly sharing data—e.g. participant privacy or use of data from a third party—those must be specified.

Reviewer #1: No

Reviewer #3: No

5. Is the manuscript presented in an intelligible fashion and written in standard English?

Reviewer #1: Yes

Reviewer #3: Yes

Reviewer #1: In this revision, the authors have addressed most of the concerns raised in the previous round, and the quality of the manuscript has greatly improved. However, several concerns remain:

1. For some reason, the Supplementary Materials mentioned in the response and the revision were not found in the compiled pdf proof (which could either be due to issues of submission system or mistakes during the submission process). This include all supplementary figures and tables, in addition to the full list of questionnaire items. Additionally, the supplementary item corresponding the questionnaire should also be explicitly cited in the main text.

2. I appreciate the authors for clarifying the reasons behind the findings related to the relevance and suitability of TM for acute events. However, I believe this point could benefit from further clarification in the text.

The authors suggest that the apparent contradiction is attributable to clinicians perceiving specific subtypes of acute situations as appropriate for TM (e.g., diagnostic support or initial triage) while regarding other acute scenarios as less suitable (e.g., those requiring direct physical intervention). It would be helpful to know whether this is a hypothesis proposed by the authors or actual observations from the experimental results, as I could not find mention of these nuances in the specific acute scenarios (and the questionnaire was not included in this revision). Were these specific acute scenarios explicitly distinguished in the questionnaire? Or is further investigation needed to fully understand the perceived suitability for TM across these scenarios? In either case, I would recommend addressing this distinction clearly in the manuscript. Since the greatest value of this work lies in its actionable insights, clarifying this context will prevent readers from being confused by the same observation. (I also noticed that the explanation provided in the response has not yet been fully incorporated into the manuscript text).

3. The survey data is currently available as a supplemental file in this submission, but the authors stated otherwise in the Data availability Statement and the response to another reviewer. Please kindly confirm whether sharing of survey data is indeed intended.

4. Typos: Introduction, “….where multiple interdependent factors are at play (17) or ML, we applied XGBoost…”

Reviewer #3: (No Response)

**Do you want your identity to be public for this peer review?** For information about this choice, including consent withdrawal, please see our Privacy Policy..

Reviewer #1: **Yes:** Shih-Yen LinShih-Yen LinShih-Yen LinShih-Yen Lin

Reviewer #3: **Yes:** Mohammad Ali TakallouMohammad Ali TakallouMohammad Ali TakallouMohammad Ali Takallou

**Figure resubmission:** While revising your submission, we strongly recommend that you use PLOS’s NAAS tool (https://ngplosjournals.pagemajik.ai/artanalysis) to test your figure files. NAAS can convert your figure files to the TIFF file type and meet basic requirements (such as print size, resolution), or provide you with a report on issues that do not meet our requirements and that NAAS cannot fix.

**Reproducibility:** To enhance the reproducibility of your results, we recommend that authors of applicable studies deposit laboratory protocols in protocols.io, where a protocol can be assigned its own identifier (DOI) such that it can be cited independently in the future. Additionally, PLOS ONE offers an option to publish peer-reviewed clinical study protocols. Read more information on sharing protocols at https://plos.org/protocols?utm_medium=editorial-email&utm_source=authorletters&utm_campaign=protocols To enhance the reproducibility of your results, we recommend that authors of applicable studies deposit laboratory protocols in protocols.io, where a protocol can be assigned its own identifier (DOI) such that it can be cited independently in the future. Additionally, PLOS ONE offers an option to publish peer-reviewed clinical study protocols. Read more information on sharing protocols at https://plos.org/protocols?utm_medium=editorial-email&utm_source=authorletters&utm_campaign=protocols

---

## [Editor Report · Decision Letter 2]

29 Mar 2026

Telemedicine adoption in cardiology: determinants and predictors identified using Bayesian model averaging and machine learning

PDIG-D-25-00449R2

Dear Dr. Muehlensiepen,

We are pleased to inform you that your manuscript 'Telemedicine adoption in cardiology: determinants and predictors identified using Bayesian model averaging and machine learning' has been provisionally accepted for publication in PLOS Digital Health.

Best regards,

Mathew V. Kiang, PhD

Section Editor

PLOS Digital Health